# Upregulation of COX4-2 via HIF-1α in Mitochondrial COX4-1 Deficiency

**DOI:** 10.3390/cells10020452

**Published:** 2021-02-20

**Authors:** Liza Douiev, Chaya Miller, Shmuel Ruppo, Hadar Benyamini, Bassam Abu-Libdeh, Ann Saada

**Affiliations:** 1Department of Genetics, Hadassah Medical Center, Jerusalem 9112001, Israel; chaya.miller7@gmail.com; 2Info-CORE, I-CORE Bioinformatics Unit of the Hebrew, University of Jerusalem and Hadassah Medical Center, Jerusalem 91120011, Israel; shmuel.ruppo@gmail.com (S.R.); hadar.benyamini@gmail.com (H.B.); 3Department of Pediatrics and Genetics, Makassed Hospital and Al-Quds University, East Jerusalem 91220, Palestine; drbassam@hotmail.com; 4Faculty of Medicine, Hebrew University of Jerusalem, Jerusalem 9112001, Israel

**Keywords:** mitochondria, cytochrome *c* oxidase, COX4-1, COX4-2, HIF-1α

## Abstract

Cytochrome-*c*-oxidase (COX) subunit 4 (COX4) plays important roles in the function, assembly and regulation of COX (mitochondrial respiratory complex 4), the terminal electron acceptor of the oxidative phosphorylation (OXPHOS) system. The principal COX4 isoform, COX4-1, is expressed in all tissues, whereas COX4-2 is mainly expressed in the lungs, or under hypoxia and other stress conditions. We have previously described a patient with a COX4-1 defect with a relatively mild presentation compared to other primary COX deficiencies, and hypothesized that this could be the result of a compensatory upregulation of COX4-2. To this end, COX4-1 was downregulated by shRNAs in human foreskin fibroblasts (HFF) and compared to the patient’s cells. COX4-1, COX4-2 and HIF-1α were detected by immunocytochemistry. The mRNA transcripts of both COX4 isoforms and HIF-1 target genes were quantified by RT-qPCR. COX activity and OXPHOS function were measured by enzymatic and oxygen consumption assays, respectively. Pathways were analyzed by CEL-Seq2 and by RT-qPCR. We demonstrated elevated COX4-2 levels in the COX4-1-deficient cells, with a concomitant HIF-1α stabilization, nuclear localization and upregulation of the hypoxia and glycolysis pathways. We suggest that COX4-2 and HIF-1α are upregulated also in normoxia as a compensatory mechanism in COX4-1 deficiency.

## 1. Introduction

The mammalian cytochrome *c* oxidase (COX, mitochondrial respiratory chain complex IV) is a dimeric multisubunit complex that is comprised of fourteen mitochondrial and nuclear-encoded subunits. COX is considered the rate-limiting complex of the oxidative phosphorylation system (OXPHOS). The two unique regulatory mechanisms (compared to the other OXPHOS complexes) are the expression of isoforms and the binding of specific regulatory factors to nuclear-encoded subunits [1,2]. One important regulatory mechanism involves the COX4 isoforms and feedback inhibition by ATP. COX4 is the largest nuclear-encoded subunit and plays important roles in COX function, assembly and regulation. It is allosterically inhibited at high ATP/ADP ratios, and by phosphorylation, allowing the fine tuning of the mitochondrial respiratory capacity. COX4 isoform 1 (COX4-1), which is the main subunit 4 of COX, is ubiquitously expressed in mammalian tissues under normoxic conditions. COX4 isoform 2 (COX4-2), the less common isoform, is primarily expressed in the lung and at lower levels in the placenta, heart, brain and pancreas. COX4-2 is preferentially expressed under hypoxia and oxidative stress, and it is suggested that the isoform switch results in a more efficient COX activity. [2,3].

In 2017, we reported a novel form of isolated COX deficiency caused by a K101N variant of the *COX4I1* gene encoding COX4-1 in a patient presenting with Fanconi anemia-like features, short stature and mild dysmorphism, while all the known Fanconi anemia (FA) gene sequences were intact. We proved the pathogenicity of the homozygous K101N variant by demonstrating 25% residual COX activity in the patient fibroblast homogenate and almost undetectable COX4-1 protein in mitochondria, and by performing complementation studies with the wild-type gene [4]. Additionally, we verified chromosomal instability by phospho-histone H2A.X (γH2AX) staining [5]. Recently, an additional pathogenic COX*4I1* variant (P152T)*,* associated with a much more severe phenotype, resembling Leigh syndrome with developmental regression, abnormal MRI and 16% residual COX activity in muscle, was reported by Pilali et al. [6].

Notably, the phenotype in our patient was different and relatively mild compared to the COX4-1 P152T patient and other patients with isolated COX deficiencies and nuclear-encoded mitochondrial diseases in general [7,8,9]. Accordingly, we hypothesized that a compensatory isoform switching to COX4-2 is able to, at least partially, compensate for the deficient COX4-1. In this work, we demonstrated elevated COX4-2 in the K101N patient’s fibroblasts as well as in a human foreskin fibroblast cell line (HFF) with stable COX4-1 knockdown. This upregulation is linked to HIF-1α stabilization, nuclear localization and the upregulation of both hypoxia and glycolysis pathways.

## 2. Materials and Methods

### 2.1. Tissue Cultures

Previously established skin primary fibroblast cultures from the patient (informed consent was obtained, IRB #0485-09) [4], controls and human foreskin fibroblasts (HFF-1 ATCC) were maintained in high-glucose DMEM supplemented with 15% fetal bovine serum, L-glutamine, pyruvate and 50 µg/mL uridine (Biological Industries, Beit Ha’emek, Israel). For immunocytochemistry, cells were seeded on µ-slide 8 well ibiTreat sterile tissue culture slides (NBT; New Biotechnology Ltd., Jerusalem, Israel). For Western blotting or RNA analysis, cells were seeded, and grown in tissue culture bottles. For positive hypoxia controls in the immunostaining and Western blotting, cells were preincubated with 500 µL of CoCl_2_ supplemented with a new fresh medium, for 6 or 24 h, at 37 °C and 5% CO_2_ [10]. All the cells were incubated at 37 °C in a humidified 5% CO_2_ atmosphere.

### 2.2. RNA Interference

For the downregulation of *COX4I1*, we employed the MISSION^®^ shRNA plasmid DNA vector system. The system included five plasmids; each targeted against a different site of *COX4I1*. In addition, we used a nonmammalian *shRNA* Control Plasmid DNA target as a control vector (Sigma-Aldrich, St.Louis, MO, USA)). We introduced each of the DNA plasmids into HEK293FT cells by co-transfection with pLP1, pLP2, and pLP/VSVG plasmids using Lipofectamine (ViraPower; Invitrogen, Carlsbad, CA, USA). Human foreskin fibroblasts were infected with viral supernatant containing polybrene. Stably transfected cells were selected with puromycin (2 µg/mL) for three weeks. Preliminary results showed that shRNA #TRCN0000232554 (COX4-1 shRNA) was the most suitable for further study.

### 2.3. Quantitative Reverse Transcription Polymerase Chain Reaction (RT-qPCR)

Total RNA was isolated from patient, HFF-shRNA, HFF-CV and healthy control primary fibroblasts with Tri-Reagent (Telron, Israel), and cDNA from poly(A)+mRNA was generated using Improm II, Promega, Madison, WI, USA. Real-time, quantitative PCR for the quantification of *COX4I1*, *COX4I2*, *PDK1*, *SLC2A1*, *HK1*, *HK2*, *GUSB* and *GAPDH* transcripts was performed using Fast SYBR GreenMaster Mix and the ABI PRISM7900HT sequence detection system (Applied Biosystems, Foster City, CA, USA). The primer sequences were used for qPCR are detailed in Table 1.

### 2.4. COX Enzymatic Activity

COX activity in cells disrupted by sonication and solubilization with 0.5 mg/mL dodecyl-β-D-maltoside (Calbiochem, San-Diego, CA, US) on ice was determined by spectrophotometry, monitoring the oxidation of 50 µM reduced cytochrome *c* at 550 nm in 10 mM potassium phosphate buffer, pH 7.0, at 37 °C on a Kontron UVICON xs double beam spectrophotometer (Secomam, Ales, France).

### 2.5. Oxygen Consumption Rate

The oxygen consumption rate (OCR) was measured using an Agilent Seahorse XF Cell Mito Stress Test Kit (#103015-100) (Seahorse Biosciences, North Billeric, MA, USA). A total of 10,000 cells/well were seeded on an XF96-well plate. Following 72 h, the growth medium was changed to an unbuffered DMEM (Agilent Technologies, Inc. Wilmington, NC, USA) supplemented with 10 mM glucose, 1 mM pyruvate and 2 mM glutamine, and the plate was equilibrated at 37 °C for 1 h before the measurements. In the absence and in the presence of sequentially added 2.5 µM oligomycin, 2 µM carbonyl cyanide-4 (trifluoromethoxy) phenylhydrazone (FCCP), 0.5 µM rotenone and antimycin, the OCR was determinedaccording to the manufacturer’s instructions. The oxygen consumption rate (OCR) and extracellular acidification rate (ECAR) were calculated relative to the cell content per well, estimated by the methylene blue assay, which is proportional to the cell count, as we have previously described [11]. Briefly, cells were fixed with 0.5% glutaraldehyde for 10 min, rinsed with double-distilled water, stained with 1% methylene blue in 0.1 M borate buffer (pH 8.5) for 1 h, rinsed with water, and allowed to dry. The dye was extracted from the cells with 0.1 N HCl at 37 °C for 1 h and then measured at A620 nm.

### 2.6. Immunofluorescence Staining

The cells were seeded on µ-slide 8 well ibi-Treat sterile tissue culture slides. On the following day, the medium was replaced by fresh medium in the presence or absence of cobalt. Following 6 h of incubation, the cells were fixed with 4% paraformaldehyde for 15 min at room temperature, and then permeabilized with either 0.3% (HIF-1 and COX4-2 staining) or 0.2% (COX4 isoform staining) Triton X-100. After blocking with 1% BSA/PBS for 1 h at room temperature, the slides were incubated with primary antibody at 4 °C overnight. On the following day, the cells were washed 3 times with PBS and incubated with fluorescence-conjugated secondary antibodies for 1 h at room temperature in the dark. The following primary antibodies were used for immunofluorescence: HIF-1α (1:500; GTX127309 GeneTex Inc, Irvine, CA USA), COX4-1 (1:150; 6431 MitoSciences, Eugene, OR, USA; A-Molecular Probes, Eugene, OR, USA) and COX4-2 (1:150; H00084701-M01, Abnova, Taipei, Taiwan). Secondary antibodies: anti-Rabbit Cy5 (711-175-152) and anti-mouse DyLight 488 (115-485-062) (both from Jackson ImmunoResearch Laboratories, Baltimore Pike, PA, USA). The slides were subsequently washed 3 times with PBS, and the nuclei were stained with Hoechst 33342, NucBlue live cell stain (Molecular Probes, Life Technologies, Eugene, OR, USA). The mitochondria were visualized using MitoTracker Red CM-H2XRos (MTR) (Molecular Probes, Life Technologies, Eugene, OR, USA). In brief, the cells were incubated with 2 µM MTR for 30 min, at 37 °C and 5% CO_2_. Then, the cells were washed with PBS and incubated with fresh high-glucose medium for another 45 min (in the dark). The cells, were then fixed as mentioned above. The cells were examined by fluorescence confocal microscopy, at ×60 or ×40 magnifications (Nikon A1R). Image analyses were performed by the quantification of fluorescence signals per nucleus (HIF-1α) or per cell (COX4 isoforms) using the Image J 1.50iJava 1.8.0_77 software (http://imagej.nih.gov/ij) (National Institute of Health, Bethesda, MD, USA).

### 2.7. Western Blotting

The Western blot assay (WB) was performed using the BIO-RAD kit. The samples were prepared as follows: the cells were washed with ice-cold PBS, harvested on ice, and immediately incubated in Laemmli sample buffer at 95 °C for 5 min. Subsequently, the samples were subjected to SDS-PAGE (Mini-Protean, any-kD precast gel), transferred to a PVDF membrane, and blocked with Every Blot Blocking Buffer according to the manufacturer’s instructions (Bio-Rad, Hercules, CA, USA). The membrane was incubated with rabbit anti-HIF-1α antibodies (GenTex, Zeeland, MI, USA; Cat#: GTX127309) (1:1000), overnight, at 4 °C. The following day, the membranes were washed with TBS-Tween and incubated with peroxidase-conjugated donkey anti-rabbit antibody (Jackson ImmunoResearch Laboratories, West Grove, PA, USA; Cat#: 111-035-144) for 1 h, at room temperature. The membranes were visualized with enhanced Clarity Max ECL detection (Bio-Rad) and were analyzed with the Fusion Solo system (Vilber Lourmat). Thereafter, the membranes were washed and then incubated with mouse monoclonal antibody against actin (1:500; Cat#: 691001, MP Biomedicals, Solon, OH, USA) for 2 h at RT, washed and incubated with peroxidase-conjugated goat-anti-mouse antibody (Cat#: 115-035-062, Jackson ImmunoResearch Laboratories, West Grove, PA, USA) for 1 h at RT and visualized as above. The band intensities were measured using ImageJ (http://imagej.nih.gov/ij) (National Institute of Health, Bethesda, MD, USA).

### 2.8. Expression by Linear Amplification and Sequencing: CEL-Seq2

Total RNA was isolated from *COX4I1* shRNA-transfected cells or control HFF. Qualification and quantification were performed using a Tapestation. A HiSeq assay using the CEL-Seq2 method [12] was performed at the Technion Genome Center, Haifa, Israel. Briefly, 3′ cDNA was synthesized and barcoded, followed by RNA synthesis and amplification by in vitro transcription.

Statistical and bioinformatic analysis of the CEL-Seq2 data was performed in collaboration with the Info-CORE Bioinformatics Unit (Hebrew University of Jerusalem and Hadassah Medical Center). Following demultiplexing, the reads were quality filtered and trimmed for adapters as well as for poly-A tails using Cutadapt [13]. Then, they were aligned to the human genome (GRCh38 with annotations from Ensembl release 95) using Tophat v2.1.1 and quantified with HTSeq [14].

Differential expression analysis was performed with the DESeq2 package (v1.22.1). Genes with a sum of counts less than 10 over all the samples were filtered out. Differential expression was calculated with default parameters except not using the independent Filtering algorithm. The significance threshold was set as FDR < 0.1.

Visualization was performed using Glimma (1.10.1) [14]. The results were combined with gene details (such as symbols, known transcripts, etc.), taken from the results of a BioMart query (Ensembl, release 95), to produce the final Excel file. In order to identify biological functions that were expected to be influenced (to either increase or decrease) given the observed gene expression changes (between HFF-shCOX4I1 and HFF), we ran gene set enrichment analysis (GSEA, reference: https://www.ncbi.nlm.nih.gov/pmc/articles/PMC1239896/). GSEA uses whole differential expression data (cut-off independent) to determine whether a priori-defined sets of genes show statistically significant, concordant differences between two biological states. We used the hallmark gene set collection from the molecular signatures database (MsigDB). The expression data results (GSE166429) are available on the GEO (Gene Expression Omnibus) website (https://www.ncbi.nlm.nih.gov/geo/) (accessed on 20 February 2021).

### 2.9. Statistical Analysis

Statistical analysis was performed by two-tailed Student’s unpaired *t*-tests using the IBM SPSS statistics for Windows, version 24.0. software (IBM Corp. Armonk, NY, USA). *p* values < 0.05 were considered statistically significant.

## 3. Results

### 3.1. COX4 Isoform Switch and Altered Energetic Profile in COX4-1-Deficient Cells

In order to characterize COX4-1 deficiency in both HFF-shCOX4I1 and in the patient’s cells, we initially quantified *COX4I1* mRNA transcripts by RT-qPCR and detected a significant 86% and 60% decrease relative to the controls, respectively. This verified our previous preliminary findings in the patient’s cells and the efficacy of the shRNA, which downregulated but did not deplete *COX4I1* expression. Reciprocally, COX4I2 transcripts were elevated 1.72 times in the HFF-shCOX4I1 cells and even more so (16 times) in the patient’s cells (Figure 1A,B). The enzymatic activity of COX was, as expected, significantly reduced, but not abolished, in both HFF-shCOX4I1 and the patient’s cells (80% and 45% decrease, respectively) (Figure 1C,D), relative to the corresponding controls. To address whether the isoform switch translates into a change in allosteric properties as was previously reported by Arnold and Kadenbach [15], we measured the COX activity in isolated mitochondria in the presence of 10 mM ATP. The COX activity (31 nmol/min/mg) in the patient’s fibroblasts’ mitochondria was not affected by ATP, while the mitochondria that were isolated from the control fibroblasts showed a significant 52% inhibition upon preincubation with ATP (115 ± 39 and 55 ± 13 nmol/min/mg, respectively). Regretfully, we could not repeat this experiment in the HFF-shCOX4I1 cells because of impaired cell growth (manuscript in preparation), which hindered mitochondrial isolation.

We also monitored both the mitochondrial respiration (indicated as the oxygen consumption rate, OCR) and the glycolysis (estimated from the extracellular acidification rate (ECAR) of the surrounding media) (Figure 1E–J). The maximal OCR reflects the maximal respiratory capacity following the addition of the uncoupler FCCP. We calculated the ATP-linked respiration, by subtracting the OCR values obtained following the addition of oligomycin (an ATP synthase inhibitor) from the basal OCR (Figure 1G,H). The background oxygen consumption in the presence of the respiratory chain inhibitors rotenone and antimycin was subtracted from the OCR values, and all the values were normalized to cell content.

When comparing the HFF-shCOX4I1 cells and the patient’s cells relative to their corresponding controls, we clearly detected different energetic profiles (Figure 1I,H). The basal and maximal OCRs and ATP-dependent OCRs in the HFF-shCOX4I1 cells were decreased by 30–40%, but to a lesser extent than expected from the *COX4I1* transcripts (Figure 1A,B). The OCRs were even less affected in the patient’s cells (Figure 1E,F). The ECAR was elevated in HFF-shCOX4I1 (8.28 versus CV 5.6 mpH/min/A620) and in the patient’s cells (5.61 versus control at 4.53 mpH/min/A620). The energy maps, combining decreased basal OCRs with increased ECARs, clearly showed a tendency towards favoring glycolysis over OXPHOS in both the HFF-shCOX4I1 and, to a somewhat less degree, the patient’s cells (Figure 1I–J).

The relative mild decrease in both OCR and COX activity in the patient’s cells and the significant residual OCR and COX activity in the HFF-shCOX4I1 cells could possibly be explained by a compensatory upregulation of *COX4I2* expression as we observed by RT-qPCR. Therefore, for verification, we analyzed the presence of the COX4-2 protein by immunofluorescence. As depicted in Figure 2A,B, the COX4-1-deficient cells indeed showed a significant reduction in the COX4-1 protein, while displaying significant reciprocal elevation in COX4-2 protein, in accord with our assumption. (Regretfully, in our hands, the anti-COX4-2 antibodies were not suitable for immunoblotting in cell homogenates.)

### 3.2. Upregulation of Glycolytic and Hypoxia Pathways in COX4-1-Deficient Cells

COX4 isoform 2 is mainly expressed in the lungs, and at low levels in the placenta, heart, brain and pancreas. The COX4 isoform 2 gene (*COX4I2*) features an oxygen responsive element (ORE), which is HIF-1-independent. COX4-2 expression is controlled by, at least, two mechanisms. That occurs either through the regulation of HIF-1 or via its ORE, where transcription is regulated by the interplay of three regulatory transcription factors: RBPJ, CXXC5 and CHCHD2 [16]. In order to identify the possible pathways that regulate the observed isoform switch, we performed CEL-Seq2 analysis of HFF-shCOX4I1 and compared the RNA expression pattern with the HFF control (with the same genetic background) from five RNA extractions performed on different occasions. The main significantly upregulated pathways were epithelial–mesenchymal transition (EMT), glycolysis and hypoxia (Figure 3A) (a list of the upregulated genes is available in the Appendix A). The upregulation of glycolysis was in accord with the previously mentioned energy map showing decreased OCRs with elevated ECARs. However, the pathway that caught our attention was hypoxia, and among the potential mediators of hypoxia-induced EMT is the hypoxia-inducible factor-1α (HIF-1α), which is a transcription factor [17] linked, among many other functions, to the upregulation of COX4-2 under hypoxia [18]. In order to validate these results, and to confirm the involvement of the HIF-1α pathway, we performed RT-qPCR validations, to detect and quantify the levels of several major HIF-1 target genes. As represented in Figure 3C, we verified the upregulation of four HIF-1α target genes (including *PDK1*, *GLUT1*, *HK1* and *HK2*) observed in the CEL-Seq2 (represented in the volcano plot: Figure 3B), which were also upregulated in both knockdown and in the patient’s fibroblasts.

### 3.3. HIF-1α Is Elevated and Translocated to the Nucleus in COX4-1-Deficient Cells

To gain further insight into the HIF-1α pathway in COX4-1 deficiency, we set out to detect the presence of HIF-1α protein in our model system.

Notably, HIF-1α is stabilized under hypoxic conditions, enabling it to be translocated to the nucleus, where it functions as a transcriptional activator. Previous studies by Fukuda et al. [18] and others showed that reduced levels of oxygen lead to the elevation of COX4-2 expression. Moreover, they claimed that COX4-2, but not COX4-1, mRNA expression levels were elevated when cells were treated with the hypoxia inducer cobalt chloride [18,19]. In order to understand whether HIF-1 plays a role in the COX4 isoform switch in COX4-1-deficient cells grown under normoxic conditions, we initially analyzed the presence of HIF-1α in whole-cell extracts by immunoblot analysis (Western blotting), showing that, indeed, the level of HIF-1α in both HFF-shCOX4I1 cell and patient’s cell lysates was markedly elevated (3.5 and 30 times, respectively) when normalized to beta-actin, relative to controls (Figure 4A,B). To validate HIF-1α activation and band migration, we added cobalt 24 h before performing the assay.

In order to confirm the presence of HIF-1α and study its cellular localization, we co-immunostained HIF-1α with COX4-2 and counterstained with Hoechst-33342. As depicted in (Figure 5), the COX4-1-deficient cells displayed increased levels of HIF-1α localized in their nuclei, indicating that the elevated level- of HIF-1α is most probably due to its stabilization. Notably, all the COX4-2-positive cells showed nuclear HIF-1α staining. The accumulation and translocation of HIF-1α to the nucleus indicate that HIF-1 was activated and thereby inducing the HIF-1 signaling pathway. These results affirm the CEL-Seq2 data analysis (Figure 3A) and supporting the idea of regulation of COX4-2 by HIF-1α.

Nevertheless, we were still puzzled by the fact that upregulation of COX4-2 and HIF-1α occurs under normoxic conditions in our system. Thus, we aimed to reinforce our hypothesis that the upregulation of COX4-2 via HIF-1α also occurs in normoxia. To this end, we compared the levels of double-stained COX4-2 and HIF-1α in untreated cells and in cells treated with cobalt, in order to mimic hypoxia inducing a chemical upregulation of COX4-2. As depicted in Figure 6A,B, 6 h of preincubation with cobalt led to HIF-1α stabilization (the accumulation of HIF-1α in the nuclei) in each type of cell line (both COX4-1-deficient cells and corresponding controls). Interestingly, while a significant increase in the levels of COX4-2 was observed in the controls, the COX4-1-deficient cells showed comparable levels of COX4-2, with and without cobalt. With respect to the nuclear accumulation of HIF-1α, a significant increase was evident upon cobalt treatment in all the cells, also in the COX4-1-deficient cells. Taken together, we suggest that the levels of COX4-2 in COX4-1-deficient cells are already, a priori, elevated to a maximum at relatively low levels of HIF-1α (compared to cobalt) (Figure 6B).

## 4. Discussion

In this work, we show that COX4-1 deficiency due to the K101N variant still leaves a significant amount of OXPHOS capacity in the patient’ fibroblasts. This was also confirmed in normal fibroblasts where COX4-1 was downregulated by shRNA. We chose downregulation in the fibroblasts over knockout since it more accurately simulates the phenotype of the patient, which is much milder than other nuclear-encoded isolated COX deficiencies [6,7,8,9]. The only mild decrease in both the basal and ATP-linked OCRs in the COX4-1 knockdown cell line, together with the comparable ATP-linked OCRs in the patient’s and control fibroblasts, is in accord with our previous results, showing relatively normal ATP production by the luciferin–luciferase assay in the patient’s cells [20]. These results suggest a cellular compensatory mechanism, for COX4-1 deficiency, in which the expression of COX4-2 is upregulated. Indeed, we detected an upregulation of the second isoform COX4-2, which, likely, is the cause of the observed partially rescued phenotype of the COX4-1-deficient cells. Since we demonstrated similar results in both the patient’s and the COX4-1-downregulated cells, we suggest that this phenomenon is not attributable solely to the K101N variant. Our results are in accord with previously published data obtained in various cell lines [17,18,19]. Recently, Reguera *et al*. studied each COX4 isoform separately by constructing HEK293-based cell lines with Cas9-mediated COX4 isoform knockout, followed by the stable knock-in of either isoform. In their study, the researchers confirmed different COX kinetics depending on which isoform (either COX4-1 or COX4-2) is expressed [21]. We also confirmed this, as COX activity in the patient’s fibroblast mitochondria was not inhibited by ATP, a feature of COX4-2. This in accord with published data showing that the allosteric ATP binding site in COX4-1 is dependent on the phosphorylation of S58, while this regulatory residue does not exist in COX4-2 [22]. Regretfully, we could not perform more in-depth kinetic studies in the HFF-shCOX4I1 cells due to impaired growth and chromosomal instability, which was more pronounced than in the patient’s cells [20], hindering growth and preventing mitochondrial isolation (manuscript in preparation). Regretfully, we could also not confirm a COX4-2 upregulation switch in the muscle of our patient, as there was no consent for muscle biopsy. Nevertheless, isoform switches can also occur in other normoxic systems, as exemplified by our previous observations in two rodent models; heat-acclimation-mediated cardioprotection in a rat model by mitochondrial metabolic remodeling included the upregulation of COX4-2 via HIF-1α in the heart, and additionally, we detected the upregulation of COX4-2 in brain tissue in a mouse model of prion disease [23,24]. Interestingly, in a reciprocal manner, we previously observed a compensatory upregulation of COX4-1 in COX4-2 deficiency [10]. Notably, besides COX4, isoforms exist in six additional, nuclear-encoded COX subunits (COX6B, COX7B, NDUFA4, COX6A, COX7A and COX8), which are not essential for catalytic function but may play regulatory roles. Some of these isoforms show tissue and developmental specificity (as recently reviewed by Čunátová et al.) [25]

In order to elucidate a possible pathway through which the upregulation of COX4-2 occurs, we performed RNA-seq analysis (CEL-Seq2) and analyzed the differential gene expression between HFF-shCOX4I1 and control cells, with the same nuclear background. The data were then analyzed by GSEA (gene set enrichment analysis) in order to detect up- and downregulated gene sets. Using this method, we showed that COX4-1 deficiency is accompanied by the upregulation of the hypoxia and glycolysis pathways. The upregulation of glycolytic flux was in accord with the elevated extracellular acidification rates (ECARs) and with the upregulation of several major glycolytic enzymes;hexokinase (*HK2*), pyruvate kinase (*PKM*), phosphoglycerate kinase (*PGK1*), phosphoglycerate mutase (*PGAM1*), triosephosphate isomerase (*TPI1*) and enolase (ENO2) (Appendix A), and preliminary proteomics results indicate upregulated HK2 protein. In order to affirm the involvement of HIF-1α, we analyzed its abundance and localization and also verified the upregulation of several HIF-1 target genes. Interestingly, HIF-1α was present in the nucleus under normoxia and without any obvious oxidative stress (ROS were not relatively elevated in both the patient’s cells [20] and in the knockdown cell lines (results not shown). Notably, under hypoxia, HIF-1α, together with HIF-1β, forms a stabilized HIF-1 complex, which acts as a transcription factor for a variety of genes that contain the hypoxia response elements, including the *COX4I2* gene [18]. There is evidence that mitochondrial signals, other than hypoxia and ROS, such as redox status (the NADH/NAD ratio), metabolites (TCA-intermediate oncometabolites, such as fumarate, succinate and lactate) and other noncanonical mechanisms, can imitate and evoke hypoxia-like responses and modulate metabolism, ( reviewed in [26,27]). Interestingly, HIF-1α also induces genetic instability, indicating that the regulation of DNA repair is an integral part of the hypoxic response [28]. These observations are in accordance with our previous publications regarding elevated levels of double-stranded breaks (DSBs) and genomic instability in COX4-1-deficient cells. [5,20]. Although glycolysis, HIF-1α and genomic instability are linked to cancer and invasiveness, we did not observe transformation of the HFF-shCOX4I1 or patient cells. Rather oppositely, preliminary results show that the HFF-shCOX4I1 cells reached senescence at a relatively early passage (manuscript in preparation).

To summarize, our results contribute to the elucidation of the role of COX4-1 in metabolism and to the current understanding of the pathomechanism of COX deficiency due to nuclear-encoded subunits, and we conclude that COX4-2 is upregulated and partially rescues COX4-1 deficiency through HIF-1α activation induced by a yet-to-be-characterized, noncanonical pathway.

## Figures and Tables

**Figure 1 cells-10-00452-f001:**
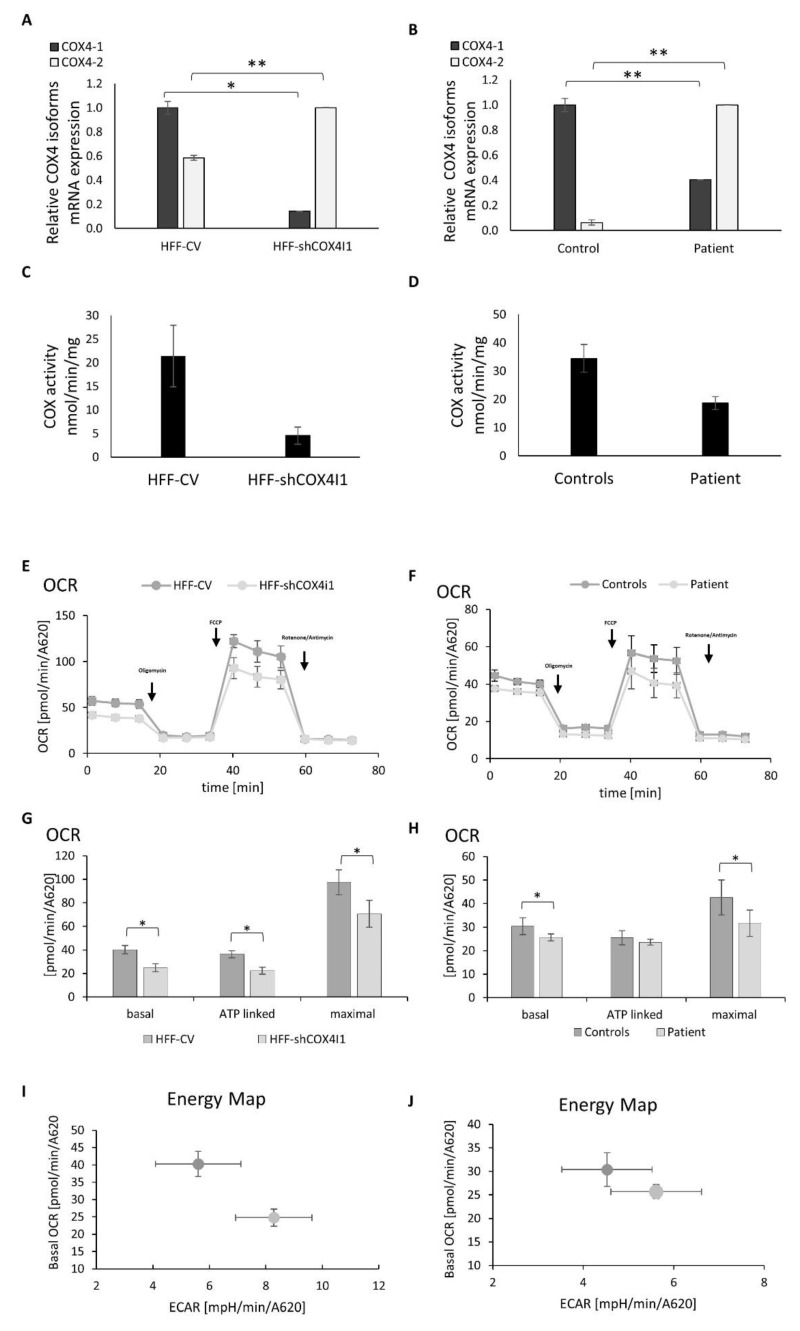
Decreased *COX4I1* expression, COX activity and mitochondrial respiration with elevated glycolysis trend in COX4-1-deficient cells. mRNA expression levels of COX4 isoforms were measured by RT-qPCR in the knockdown vs. control cell lines (COX4-1 knockdown was obtained using a stable expression of COX4I1-targeting or nonmammalian-targeting shRNAs, respectively) (**A**) and in patient’s and healthy control fibroblasts (**B**). The results reveal decreased levels of COX4I1 mRNA expression and reciprocally elevated levels of *COX4I2* mRNA expression in both COX4-1-deficient cells. Enzymatic activity of COX determined by spectrophotometry; shows decreased activity (**C**,**D**) in both shCOX4I1 and patient’s cells, relative to corresponding controls. Oxygen consumption rate (OCR) and extracellular acidification rate; extracellular acidification rates (ECARs) were measured in shCOX4I1 (**E**,**G**,**I**) and patient cells (**F**,**H**,**J**) and in corresponding controls, with subsequent addition of oligomycin, FCCP and antimycin/rotenone. Basal, ATP-linked and maximal OCR were calculated (**G**,**H**), and OCR was plotted against ECAR to construct energy maps (**I**,**J**). All values were normalized to cell content measured by methylene blue (A620). OCR was reduced, relative to the corresponding controls. Both COX4-1-deficient cell lines showed a tendency towards more glycolysis than the corresponding controls. Values are presented as normalized mean ± SEM; * *p* < 0.05 and ** *p* < 0.01 compared to corresponding controls.

**Figure 2 cells-10-00452-f002:**
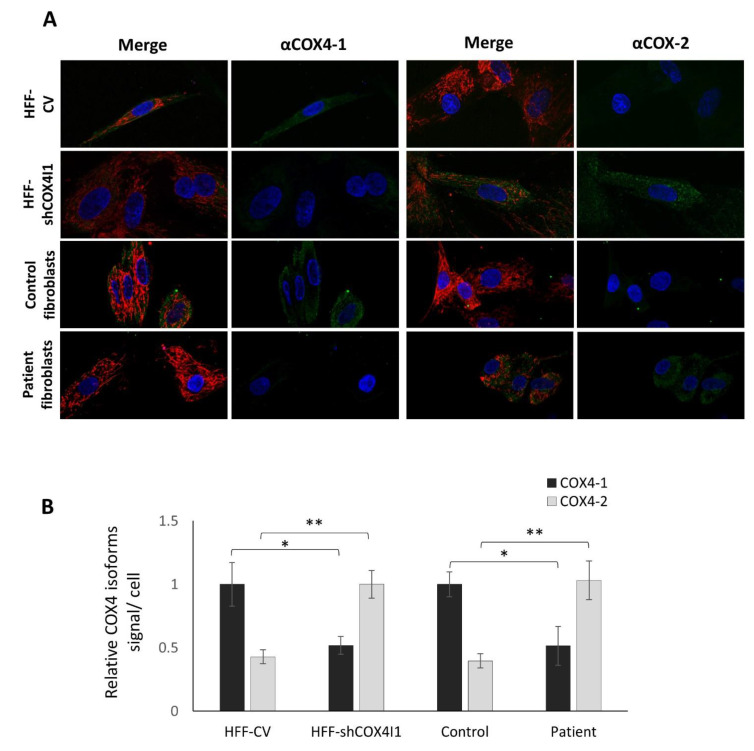
Decreased COX4-1 with a reciprocal elevation of COX4-2 in COX4-1-deficient cells. HFF-shCOX4I1, patient’s and corresponding control cells were incubated with MitoTracker red, fixed and stained separately with antibodies against COX4-1 and COX4-2. The results displayed in **A** and **B** demonstrate markedly decreased COX4-1 levels in both COX4-1-deficient cells (HFF-shCOX4I1 and patient’s cells), while COX4-2 staining was increased, relative to respective controls. Nuclei were visualized with Hoechst-33342 (A). The micrographs were quantified and are depicted as histograms of signal intensity per cell ± SEM of at least 100 cells * *p* < 0.05; ** *p* < 0.01 (**B**).

**Figure 3 cells-10-00452-f003:**
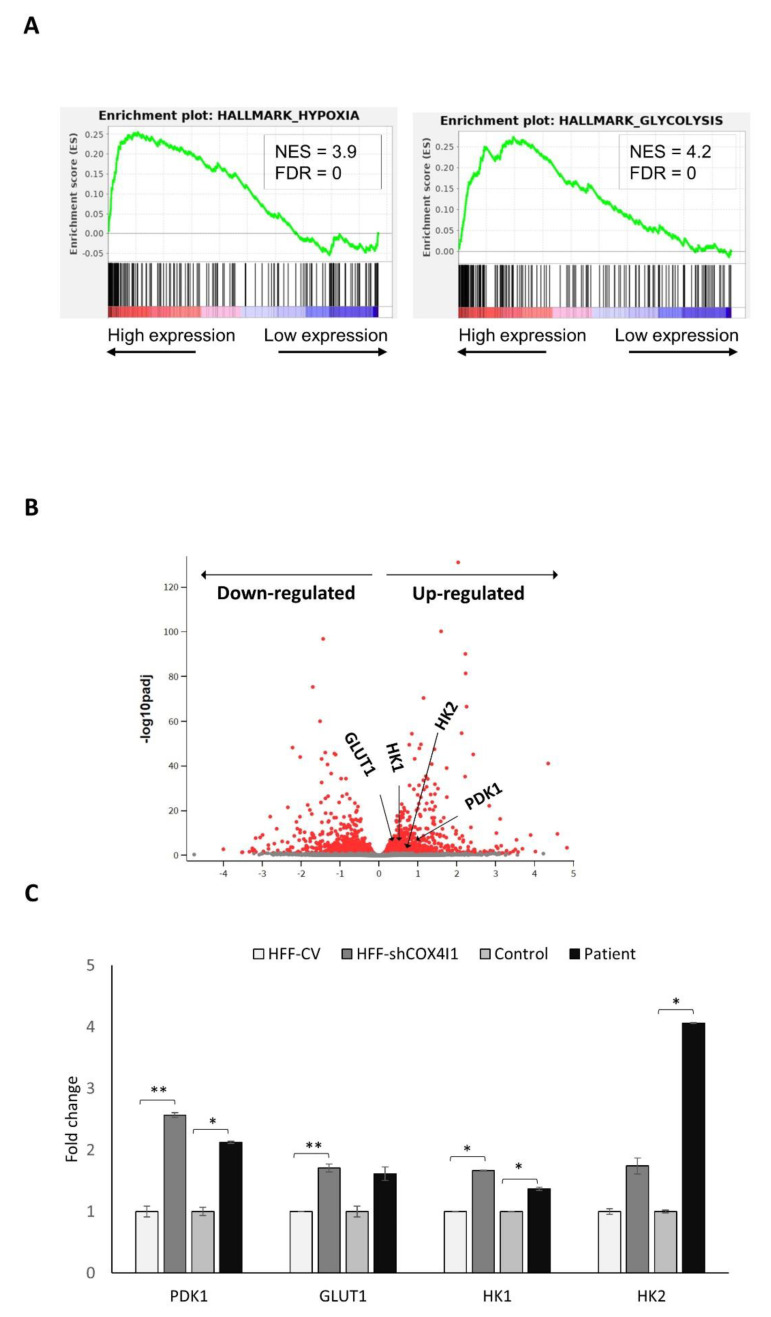
CEL-Seq2 analysis identified hypoxia as one of the top upregulated pathways in COX4-1-deficient cells. Total RNA was isolated from both COX4-1-deficient cells (HFF-shCOX4I1 and HFF) and from both types of control cells (HFF-CV and healthy control fibroblasts). A HiSeq assay was performed at the Technion Genome Center using the CEL-Seq2 method. Differential expression analysis was performed with the DESeq2 package. Significance threshold was set as FDR < 0.1. In order to identify biological functions that were expected to be influenced (either to increase/decrease) given the observed gene expression changes (between HFF-shCOX4I1 and HFF fibroblasts), pathway analysis was subsequently performed. Upregulation of hypoxia (left) and glycolysis (right) detected by gene set enrichment analysis (GSEA). NES: normalized enrichment signal; FDR: false discovery rate (**A**). In the volcano plot, each dot represents a gene (**B**). The *x*-axis indicates the log2 (fold change) of the expression of HFF-shCOX4I1 relative to healthy control fibroblasts, and the *y*-axis reflects –log10 of the FDR-adjusted *p*-value of this comparison. The colored dots pass the threshold for FDR. Selected HIF-1 target genes in the volcano plot (*PDK1*, *GLUT1*, *HK1* and *HK2*) were validated by RT-qPCR. Values of RT-qPCR validation are presented as the log2 (fold change) in ± SD of three biological duplicates, * *p* < 0.05, ** *p* < 0.01 (**C**).

**Figure 4 cells-10-00452-f004:**
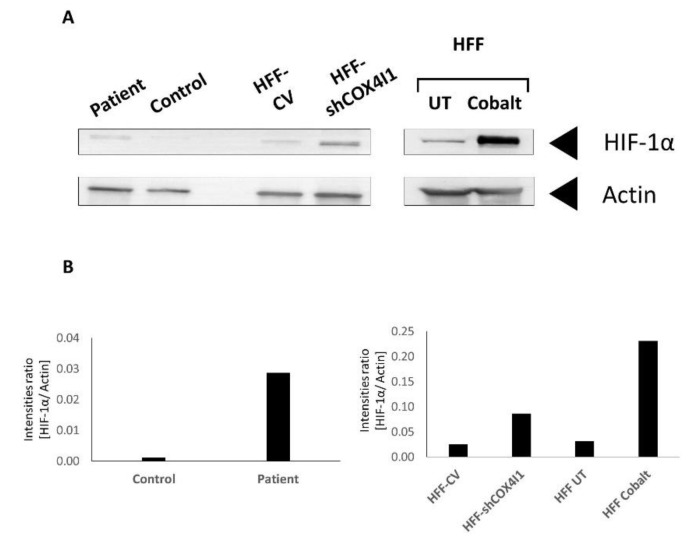
Increased levels of HIF-1α in COX4-1-deficient cells. Western blot analysis was performed on cell extracts from both COX4-1-deficient cells (patient and HFF-shCOX 1) and from both corresponding controls. The extracted cells were probed with anti-HIF-1α and anti-actin antibodies as loading control. As a positive control for HIF1-α, untreated (UT) human foreskin fibroblasts (HFF) were preincubated with cobalt chloride to simulate hypoxia (**A**). The histogram represents the results normalized to actin (**B**). The figure depicts one representative experiment out of three, showing an increased level of HIF-1α in COX4-1-deficient cells.

**Figure 5 cells-10-00452-f005:**
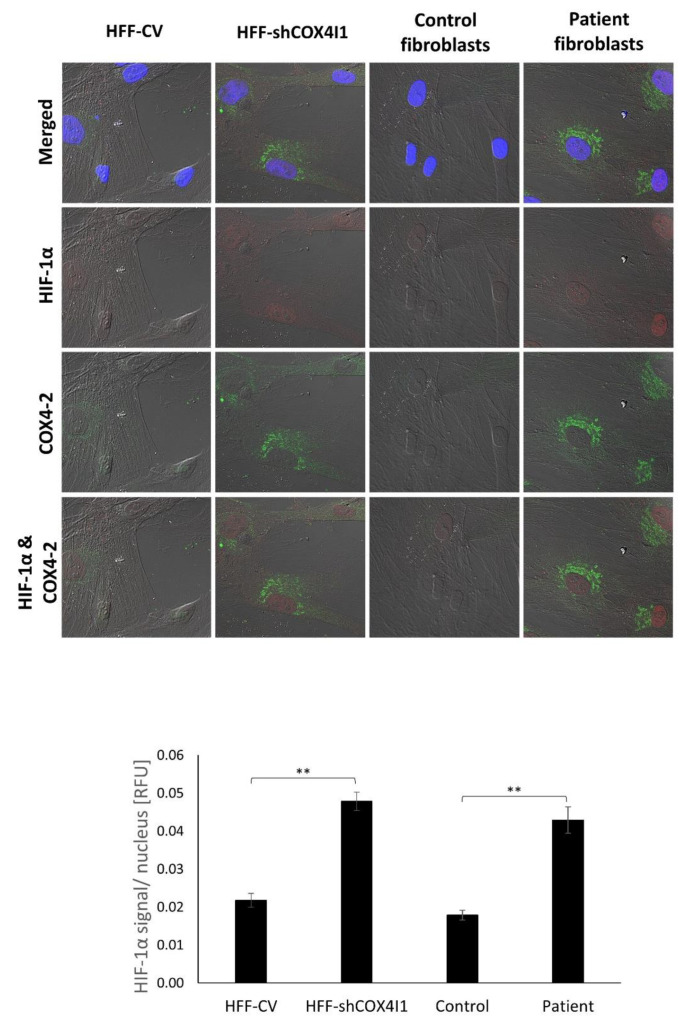
HIF-1α nuclear accumulation was present in COX4-2-positive cells. COX4-1-deficient (HFF-shCOX4I1 and patient) and control cells were stained with antibodies against HIF-1α (red) and COX4-2 (green). Nuclei were visualized by Hoechst-3334. The results demonstrate increased nuclear localization of HIF-1α in COX4-1-deficient cells relative to controls. The observed accumulation of HIF-1α was correlated with COX4-2-positive cells (upper panel). The micrographs were quantified and are depicted as histograms of HIF-1α (lower panel) relative signal intensity per nucleus ± SEM of at least 100 nuclei ** *p* < 0.01.

**Figure 6 cells-10-00452-f006:**
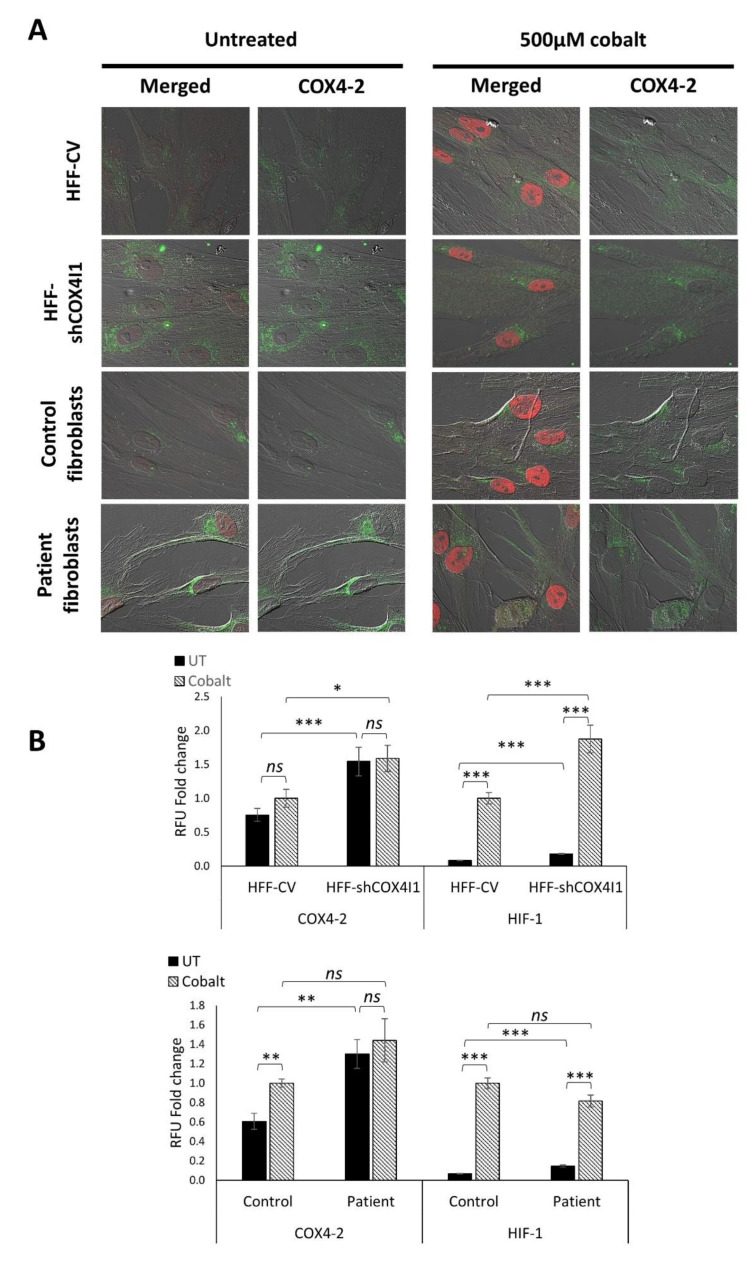
Chemical activation of HIF-1α increased COX4-2 expression in control cells, whereas in COX4-1-deficient cells, the levels remained unchanged. COX4-1-deficient (HFF-shCOX4I1 and patient) and control cells were preincubated either with or without cobalt for 6 h prior to performing co-staining of COX4-2 and HIF-1α (**A**). The stabilization of HIF-1α was demonstrated by its translocation to the nuclei of each treated cell. COX4-1-deficient cells did not show any difference in COX4-2 levels with and without cobalt treatment, while nuclear accumulation of HIF-1α was increased (**A**). The quantified results are depicted in the histogram (**B**). Values are normalized to the corresponding cobalt-treated controls. Mean ± SEM of at least 70 cells; * *p* < 0.05, ** *p* < 0.01, *** *p* < 0.001 compared to corresponding control; ns—not significant.

**Table 1 cells-10-00452-t001:** Primers qPCR

Gene	Forward Primer	Reverse Primer
COX4I1 (NM_001861.6)	5′-TTTCACCGCGCTCGTTAT-3′	5′-CTTCATGTCCAGCATCCTCTT-3′
COX4I2 (NM_032609)	5′-GAAGACGAGGGATGCACAG-3′	5′-GGCTCTTCTGGCATGGG-3′
PDK1 (NM_001278549.2)	5′- CAGGACACCATCCGTTCAAT-3′	5′- AGCTTTAGCATCCTCAGCAC-3′
GLUT1 (NM_006516.4)	5′-GCTACAACACTGGAGTCATCAA-3′	5′-ACTGAGAGGGACCAGAGC-3′
HK1 (NM_000188.3)	5′- CCCTAAATGCTGGGAAACAAAG-3′	5′- CCCTTCTTGGTGAAGTCGATTA-3′
HK2 (NM_000189.5)	5′- CATCCTCCTCAAGTGGACAAA-3′	5′- ACCACATCCAGGTCAAACTC-3′
GUSB (NM_000181.4)	5′-GAAAATATGTGGTTGGAGAGCTCATT-3′	5′-CCGAGTGAAGATCCCCTTTTTA-3′
GAPDH (NM_001357943.2)	5′-CAAGAGCACAAGAGGAAGAGAG-3′	5′-CTACATGGCAACTGTGAGGAG-3′

## Data Availability

The expression data results (GSE166429) are available on the GEO (Gene Expression Omnibus) website (https://www.ncbi.nlm.nih.gov/geo/) (accessed on 20 February 2021).

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
