# Peer review of "Upregulation of COX4-2 via HIF-1α in Mitochondrial COX4-1 Deficiency"

_cells, 2021, doi:10.3390/cells10020452_

Round 1
Reviewer 1 Report
The paper by Douiev et al., reports on in vitro studies in human fibroblasts of a previously described patient with a COX4-1 defect with a relatively mild presentation compared to other primary COX deficiencies. The authors detected elevated level of another isoform, COX4-2 in fibroblasts of this patient, an also in cells after shRNA knockdown of COX4-1. The data demonstrate elevated COX4-2 levels in the COX4-1-deficient cells with a concomitant HIF-1α stabilization, nuclear localization and upregulated hypoxia and glycolysis pathways. The authors suggest that COX4-2 and HIF-1α are upregulated as a compensatory mechanism in COX4-1 deficiency.
- I think it is a very interesting hypothesis and may be relevant more widely in nuclear COX subunit mutations. Several nuclear COX subunit genes have different tissue specific isoforms, and the role of these has not been fully understood so far. It would be useful to discuss about this possibility.
- The experiments were done in fibroblasts. Do the authors think that other tissues and cells have similar upregulation of COX4-2 to rescue COX4-1 defect? Is there any chance that they can comment on findings or discuss literature data in other tissues, such as skeletal muscle?
Author Response
Response:
- We thank the reviewer for the helpful comments and have now expanded the discussion to include additional nuclear COX subunit isoforms (reviewed by Čunátová et al, now ref 25)
- Regretfully, we could not confirm the isoform switch in muscle of this patient this as there was no consent for muscle biopsy. Nevertheless, in our previous research we reported that COX 4-2 and HIF are upregulated heart tissue upon heat adaptation in a rat model and COX 4-2 was upregulated in brain tissue of a prion mouse model. We have added this to the discussion and in this context added two references (23,24).
Reviewer 2 Report
In this manuscript authors showed that down regulation of one of the enzymes of OXPHOS Cox4-1 by ShRNA leads to up-regulation of another isoform Cox4-2. Surprisingly reported that up-regulation is positively regulated by HiF1a. All the experiments are well designed, and results are elucidated well with final discussion. In this manuscript, I would like to ask question to authors that, do they find any defects in mitochondria? Next, what is the pathological significance in general, do the cells show more invasive phenotype since they more rely on glycolysis. Need to show this point. I also curious to know whether the different major glycolytic enzyme activities up regulated. Some of the minor things authors wants to improve is immunofluorescence assay to show better images.
Author Response
We thank the reviewer for the insight and comments and made our best effort to answer the questions.
Q1: In this manuscript, I would like to ask question to authors that, do they find any defects in mitochondria?
Response to Q1: There is and energetic defect and COX deficiency. The COX defect was isolated thus we did not detect decreased activities in other respiratory chain enzymes in this patient (reference 4). We have now pointed out in the introduction that the COX defect was isolated.
Q2: Next, what is the pathological significance in general, do the cells show more invasive phenotype since they more rely on glycolysis. Need to show this point.
Response to Q2: These cells are not cancerous so we cannot relate to invasiveness. They do depend on glycolysis as many other cells with respiratory chain defects. Preliminary work (manuscript in preparation) show that the cells rather reach senescence at earlier passages.
The following sentence was added in the discussion
“Although, glycolysis, HIF-1α and genomic instability is linked to cancer and invasiveness, we did not observe transformation of the HFF-shCOX4I1 or patient cells. Rather the opposite, preliminary results show that the HFF-shCOX4I1 cells reached senescence at a relatively early passage (manuscript in preparation)”
Q3: I also curious to know whether the different major glycolytic enzyme activities up regulated
Response to Q3.
- A) The upregulation of glycolytic flux was in accord with elevated extracellular acidification rates (ECAR). (now added in the discussion).
ECAR was elevated in HFF-shCOX4I1 (8.28 versus CV 5.6 mpH/min/A620 ) and patient’s cells (5.61 versus control 4.53 mpH/min/A620 (now added in the results section)
- B) The list of upregulated genes (supplementary table) include five major glycolytic enzymes hexokinase (HK2 also verified by RT-PCR), pyruvate kinase (PKM), phosphoglycerate kinase (PGK1), Phosphoglycerate Mutase (PGAM1), Triosephosphate Isomerase (TPI1) and enolase (ENO2) and preliminary proteomics result indicate upregulation of HK2. (now added in the discussion)
Q4 Some of the minor things authors wants to improve is immunofluorescence assay to show better images.
Response to Q4.
We apologize for the inconvenience. High quality figures were uploaded in a separate zip- file.